DATA RELEASE

# The female urinary microbiota in relation to the reproductive tract microbiota

Chen Chen[1,2,5,†], Lilan Hao[1,2,†], Weixia Wei[3,4,†], Fei Li[1], Liju Song[1,2], Xiaowei Zhang[1,2], Juanjuan Dai[3,4], Zhuye Jie[1,2,6], Jiandong Li[1,2], Xiaolei Song[1], Zirong Wang[1], Zhe Zhang[1,2], Liping Zeng[3,4], Hui Du[3,4], Huiru Tang[3,4], Tao Zhang[1,2], Huanming Yang[1,7], Jian Wang[1,7], Susanne Brix[8], Karsten Kristiansen[1,5], Xun Xu[1,2], Ruifang Wu[3,4,*] and Huijue Jia[1,2,6,9,*]

1   BGI-Shenzhen, Shenzhen 518083, China
2   China National Genebank, BGI-Shenzhen, Shenzhen 518120, China
3   Peking University Shenzhen Hospital, Shenzhen 518036, China
4   Shenzhen Key Laboratory on Technology for Early Diagnosis of Major Gynecological diseases, Shenzhen, PR China
5   Department of Biology, Ole MaalØes Vej 5, University of Copenhagen, Copenhagen, Denmark
6   Shenzhen Key Laboratory of Human Commensal Microorganisms and Health Research, BGI-Shenzhen, Shenzhen, China
7   James D. Watson Institute of Genome Sciences, Hangzhou, China
8   Department of Biotechnology and Biomedicine, Technical University of Denmark, Soltofts Plads, Building 221, 2800 Kgs. Lyngby, Denmark
9   Macau University of Science and Technology, Taipa, Macau 999078, China

**Submitted:** 01 May 2020

\* Corresponding authors. E-mail: jiahuijue@genomics.cn; wurf100@126.com

† Contributed equally.

Preprint submitted at https://doi.org/10.1101/628974

## ABSTRACT

Human urine is traditionally considered to be sterile, and whether the urine harbours distinct microbial communities has been a matter of debate. Potential links between female urine and reproductive tract microbial communities is currently not clear. Here, we collected urine samples from 147 Chinese women of reproductive age and explored the nature of colonization by 16S rRNA gene amplicon sequencing, quantitative real-time PCR, and live bacteria culture. To demonstrate the utility of this approach, the intra-individual Spearman's correlation was used to explore the relationship between urine and multiple sites of the female reproductive tract. PERMANOVA was also performed to explore potential correlations between the lifestyle and various clinical factors and urinary bacterial communities. Our data demonstrated distinct bacterial communities in urine, indicative of a non-sterile environment. *Streptococcus*-dominated, *Lactobacillus*-dominated, and diverse type were the three most common urinary bacterial community types in the cohort. Detailed comparison of the urinary microbiota with multiple sites of the female reproductive tract microbiota demonstrated that the urinary microbiota were more similar to the microbiota in the cervix and uterine cavity than to those of the vagina in the same women. Our data demonstrate the potential connectivity among microbiota in the female urogenital system and provide insight and resources for exploring diseases of the urethra and genital tract.

**Subjects**  Genetics and Genomics, Metagenomics, Medical Microbiology Molecular Infection Biology

## DATA DESCRIPTION

## Purpose of data acquisition

The role of microbiota in the vaginal environment has received a lot of attention over the past decade, while the female upper reproductive tract was traditionally believed to be sterile and mostly studied in the context of infections or incontinence [1]. Despite continued controversy, the presence of microorganisms beyond the cervix (i.e. the female upper reproductive tract) is increasingly recognized even in non-infectious conditions [2]. Like the female upper reproductive tract, the sterile hypothesis of urine has also been overturned by emerging evidence that indicates the existence of microorganisms in the urinary tract by culturing or sequencing approaches [3, 4]. A recent study using an expanded quantitative urine culture in combination with whole-genome sequencing has isolated and sequenced the genomes of 149 bacterial strains from catheterized urine of both symptomatic and asymptomatic peri-menopausal women [5]. It also showed highly similar strains of commensal bacteria in both the bladder and vagina of the same individual [5]. Another study analysed the urinary microbiota of 189 individuals using 16S rRNA gene amplicon sequencing and suggested that the urethra and bladder can harbour microbial communities distinct from the vagina [6]. However, the relationship between female urine microbiota and the upper reproductive tract microbiota has so far not been studied.

Here, we present a dataset of the urinary microbiota for a relatively large cohort of 147 women of reproductive age. Together with our recently published study of peritoneal fluid, uterine, and vaginal samples from the same individuals [2], this data shows that although urinary microbiota contain larger populations of *Lactobacillus* and *Streptococcus*, they are more similar to the microbiota of the cervix and uterine cavity, in accordance with the anatomical opening of the bladder. Together with a wealth of metadata, we demonstrate that these data are useful for exploring the potential of the urinary microbiota for clinical diagnosis.

## METHODS

A protocol collection including methods for DNA extraction, bioinformatics analysis and quantitative real-time PCR is available via protocols.io (Figure 1) [7].

## Sample collection

In this study, a total of 147 reproductive age women (age 22–48) were recruited by Peking University Shenzhen Hospital [8]. All participants were reproductive age women who underwent hysteroscopy and/or laparoscopy for conditions without infections, such as hysteromyoma, adenomyosis, endometriosis, or salpingemphraxis. Subjects with other related diseases, such as vaginal inflammation, severe pelvic adhesion, endocrine or autoimmune disorders were removed. Pregnant women, breastfeeding women, and menstruating women at the time of sampling were also excluded. None of the subjects received any antibiotic treatments or vaginal medications within two weeks of sampling. In addition, no cervical treatment was performed within the previous 7 days, no vaginal douching was performed within 5 days, and no sexual activity was performed within at least 2 days.

137 self-sampling morning mid-stream urine samples were collected between December 2013 and July 2014 prior to the surgery (**sample_metadata.csv** [8]), and then stored at −80 °C until they were transported on dry ice to BGI-Shenzhen for sequencing. The samples

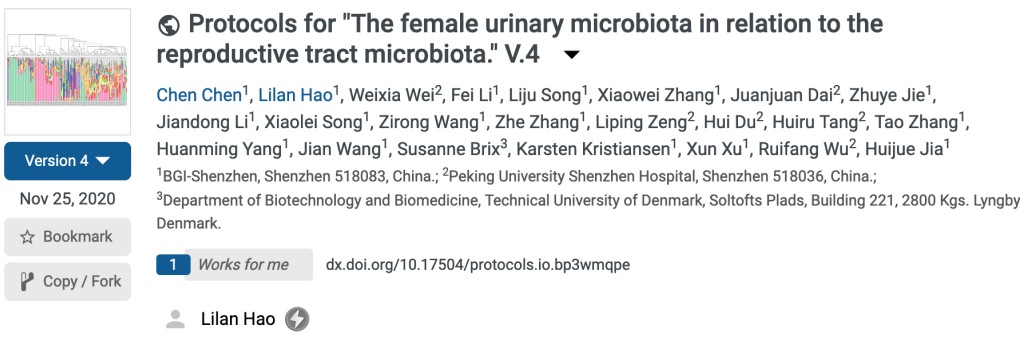

Protocols for "The female urinary microbiota in relation to the reproductive tract microbiota." V.4

Chen Chen[1], Lilan Hao[1], Weixia Wei[2], Fei Li[1], Liju Song[1], Xiaowei Zhang[1], Juanjuan Dai[2], Zhuye Jie[1], Jiandong Li[1], Xiaolei Song[1], Zirong Wang[1], Zhe Zhang[1], Liping Zeng[2], Hui Du[2], Huiru Tang[2], Tao Zhang[1], Huanming Yang[1], Jian Wang[1], Susanne Brix[3], Karsten Kristiansen[1], Xun Xu[1], Ruifang Wu[2], Huijue Jia[1]

[1]BGI-Shenzhen, Shenzhen 518083, China.; [2]Peking University Shenzhen Hospital, Shenzhen 518036, China.; [3]Department of Biotechnology and Biomedicine, Technical University of Denmark, Soltofts Plads, Building 221, 2800 Kgs. Lyngby, Denmark.

Version 4

Nov 25, 2020

☆ Bookmark

⌐ Copy / Fork

1 *Works for me*    dx.doi.org/10.17504/protocols.io.bp3wmqpe

👤 Lilan Hao ⚡

**Figure 1.** Protocol collection for sequencing and analysing female urinary microbiota. https://www.protocols.io/widgets/doi?uri=dx.doi.org/10.17504/protocols.io.bp3wmqpe

from an additional 10 women were collected for validation purposes by a doctor during the surgery in July 2017. For each operation, a urine catheter was inserted into the disinfected urethra to collect mid-stream urine. For each sample of urine collected through a catheter, an identical volume of saline solution was set as the control sample. The samples were then placed at 4 °C, transported to BGI-Shenzhen, and processed within 6 hours. A portion of each sample was used for culturing live bacteria and the rest was used for sequencing.

## DNA extraction and 16S rRNA amplicon sequencing

Genomic DNA extraction was carried out following the protocol [9]. The primers 515F and 907R were utilized for PCR amplification of the hypervariable regions V4-V5 of the bacterial 16S rRNA gene. The 907R primer includes a unique barcoded fusion. The primer sequences were: 515F: 5′-GTGCCAGCMGCCGCGGTAA-3′ and 907R: 5′-CCGTCAATTCMTTTRAGT-3′, where M denotes A or C and R denotes purine. The conditions for PCR amplification were: 3 min of denaturation at 94 °C, followed by 25 cycles of 45 s at 94 °C (denaturing), 60 s at 50 °C (annealing), and 90 s at 72 °C (elongation), followed by a final elongation for 10 min at 72 °C. The amplification products were purified by the AxyPrep™ Mag PCR Clean-Up Kit (Axygen, USA). The amplicon libraries were constructed with an Ion Plus Fragment Library Kit (Thermo Fisher Scientific Inc.) [10], then sequenced by the Ion PGM™ Sequencer with the Ion 318™ Chip v2 with a read length of 400 bp (Thermo Fisher Scientific Inc., Ion PGM™ Hi-Q™ OT2 Kit, Cat.No: A27739; Ion PGM™ Hi-Q™ Sequencing Kit, Cat.No: A25592) [11]. All experiments were performed in the laboratory of BGI-Shenzhen.

## Processing of sequencing reads

The raw sequencing reads were first subjected to Mothur (Mothur, RRID:SCR_011947; V1.33.3) [12] for filtering out the low-quality reads meeting the following criteria: (1) reads shorter than 200 bp; (2) reads not matching the degenerated PCR primers for up to two errors; (3) reads with an average quality score less than 25. A total of 8,812,607 reads, with an average of 57,225 reads per sample (a minimum of 1113 reads and a maximum of 194,564 reads) were obtained. Subsequently, the sequences with identity greater than 97% were clustered into Operational Taxonomic Units (OTUs) using the QIIME (QIIME, RRID:SCR_008249; V1.8.0) uclust programme [13], where each cluster was thought of as representing a species. The seed sequences of each OTU were aligned against the Greengene

reference sequences (gg_13_8_otus) for annotation using Mothur. The detailed analysis workflow was deposited in protocols.io [14].

We also calculated the Unifrac distance using QIIME based on taxonomic abundance profiles at the OTU level [11].

## PERMANOVA on the influence of phenotypes

Permutational multivariate analysis of variance (PERMANOVA) was used to assess the effect of different covariates based on the relative abundances of OTUs of the samples [15, 16] using Bray-Curtis and UniFrac distance and 9999 permutations from the vegan package (vegan, RRID:SCR_011950) in R [16, 17].

## Quantitative real-time PCR

We quantified the four *Lactobacillus* species, including *L.iners*, *L jensenii*, *L. crispatus* and *L. gasseri* using the modified qPCR protocol [18]. SYBR Premix Ex Taq GC (TAKARA) was used and the reactions were run on a StepOnePlus Real-time PCR System (Life Technologies). Each PCR reaction mixture contained 10 $\mu$l of 2×SYBR Premix Ex Taq GC, 0.2 $\mu$M forward primer, 0.2 $\mu$M reverse primer, 1.6 $\mu$l of DNA sample, and 8.2 $\mu$l of ultrapure water to make up the final reaction volume of 20 $\mu$l. Each run included a standard curve and all samples were amplified in triplicate. Ultrapure water was used as the blank control template.

To construct the standard curves, the sequencing-confirmed plasmids of four species were used after quantification with a Qubit Fluorometer and serial 10-fold dilutions. The amplification efficiency was (100 ±10)% and linearity values were all ≥0.99. The detailed procedure was deposited in protocols.io [19].

## Bacterial culturing

The urine samples and controls from 10 additional subjects were cultivated in the laboratory by spreading 100 $\mu$l of sample on different agars containing 5% horse blood, such as PYG agar (DSMZ 104 medium), BHI agar, and EG agar. The plates were incubated in both aerobic and anaerobic conditions at 37 ˚C for 72 hours. To keep the medium anaerobically during culture, resazurin and cysteine-HCl were added as reducing agents. The genomic DNA of the isolates was extracted by the Bacterial DNA Kit (OMEGA) and then underwent 16S rRNA gene amplification using the universal primers 27F/1492R [20]. The amplicons were purified and sent for Sanger sequencing. The generated sequences were then submitted to BLAST on the EzBioCloud [21] for identification.

## PRELIMINARY ANALYSIS AND VALIDATION
## Microbiota composition of the urine

To explore the urinary microbiota in this dataset, morning midstream urine (UR) was self-collected prior to surgery from an exploratory cohort of 137 Chinese women recruited for the study (median age 31.6, range 22–48). As with our previous vagino-uterine microbiota study [2], all volunteers had conditions that were not known to involve infections [8]. From 95 women in the cohort, six locations within the female reproductive tract, including the lower third of the vagina (CL), the posterior fornix (CU), cervical mucus (CV), endometrium (ET), left and right fallopian tubes (FLL and FRL), and peritoneal fluid (PF) were also sampled. Their vagino-uterine microbiota information have been published previously [2]. After 16S rRNA gene amplicon sequencing, the sequencing reads were



**Table 1.** Sequencing and annotation of the 137 samples from the exploratory cohort.

| Sample name | Sequencing amount | | | % of reads annotated to taxa | | Archive accession number |
|---|---|---|---|---|---|---|
| | #raw reads | #clean reads | #filtered reads | Genus | Species | |
| C001UR | 52506 | 11326 | 9347 | 100.00% | 75.51% | SAMEA5042945 |
| C002UR | 55955 | 14529 | 5930 | 100.00% | 65.36% | SAMEA5042987 |
| C003UR | 61367 | 21181 | 16831 | 100.00% | 91.06% | SAMEA5043040 |
| C004UR | 54585 | 18506 | 3325 | 100.00% | 41.59% | SAMEA5042979 |
| C005UR | 52177 | 22856 | 20683 | 100.00% | 92.62% | SAMEA5043003 |
| C007UR | 50766 | 15468 | 8737 | 100.00% | 71.10% | SAMEA5043001 |
| C008UR | 53748 | 14062 | 6169 | 100.00% | 64.05% | SAMEA5043004 |
| C009UR | 53383 | 12247 | 11327 | 100.00% | 96.97% | SAMEA5043046 |
| C011UR | 47814 | 13058 | 11292 | 100.00% | 66.76% | SAMEA5042941 |
| C012UR | 55279 | 16923 | 7484 | 100.00% | 55.46% | SAMEA5042938 |
| C014UR | 55713 | 15175 | 8818 | 100.00% | 64.31% | SAMEA5043060 |
| C016UR | 73372 | 22054 | 17669 | 100.00% | 57.04% | SAMEA5043009 |
| C018UR | 69142 | 26505 | 23581 | 100.00% | 12.12% | SAMEA5043006 |
| C019UR | 72249 | 17868 | 14440 | 100.00% | 44.63% | SAMEA5043054 |
| C020UR | 54574 | 19391 | 5452 | 100.00% | 63.83% | SAMEA5042942 |
| C021UR | 58118 | 17294 | 12123 | 100.00% | 55.33% | SAMEA5042998 |
| C023UR | 47476 | 18452 | 16795 | 100.00% | 16.09% | SAMEA5042947 |
| C026UR | 46583 | 16741 | 3267 | 100.00% | 83.96% | SAMEA5042969 |
| C028UR | 88245 | 26955 | 19268 | 100.00% | 19.93% | SAMEA5042984 |
| C033UR | 90431 | 31998 | 26496 | 100.00% | 66.99% | SAMEA5043062 |
| C035UR | 63773 | 27044 | 24115 | 100.00% | 97.31% | SAMEA5043037 |
| C038UR | 55562 | 10165 | 9208 | 100.00% | 84.56% | SAMEA5042972 |
| C039UR | 77957 | 18891 | 15748 | 100.00% | 33.38% | SAMEA5042963 |
| C040UR | 58940 | 12555 | 5438 | 100.00% | 69.49% | SAMEA5043021 |
| C041UR | 60028 | 15361 | 9366 | 100.00% | 78.38% | SAMEA5043000 |
| C042UR | 74086 | 14402 | 11088 | 100.00% | 67.53% | SAMEA5042955 |
| C043UR | 74146 | 23691 | 18730 | 100.00% | 60.19% | SAMEA5043032 |
| C045UR | 61249 | 17801 | 10367 | 100.00% | 2.96% | SAMEA5043048 |
| C047UR | 47742 | 11940 | 3506 | 100.00% | 54.25% | SAMEA5043024 |
| C048UR | 35550 | 1100 | 816 | 100.00% | 62.75% | SAMEA5042931 |
| C050UR | 51565 | 18902 | 290 | 100.00% | 72.76% | SAMEA5042936 |
| C051UR | 58783 | 10403 | 8234 | 100.00% | 50.77% | SAMEA5042983 |
| C053UR | 32311 | 1653 | 26 | 100.00% | 73.08% | SAMEA5043035 |
| C055UR | 45054 | 13326 | 6184 | 100.00% | 56.40% | SAMEA5043016 |
| C056UR | 69173 | 24652 | 8282 | 100.00% | 86.78% | SAMEA5043023 |
| C057UR | 64417 | 27033 | 24444 | 100.00% | 98.11% | SAMEA5043059 |
| C058UR | 42089 | 1415 | 912 | 100.00% | 4.28% | SAMEA5042935 |
| C059UR | 53642 | 12618 | 577 | 100.00% | 74.70% | SAMEA5042930 |
| C060UR | 73930 | 22110 | 19192 | 100.00% | 20.17% | SAMEA5043008 |
| C062UR | 63220 | 19932 | 17112 | 100.00% | 79.58% | SAMEA5043012 |
| C063UR | 44974 | 1201 | 790 | 100.00% | 53.42% | SAMEA5043039 |
| C064UR | 63505 | 15051 | 7134 | 100.00% | 82.38% | SAMEA5042981 |
| C065UR | 53884 | 15094 | 13794 | 100.00% | 52.97% | SAMEA5043027 |
| C066UR | 63269 | 16157 | 12090 | 100.00% | 45.86% | SAMEA5042985 |
| C067UR | 55812 | 19047 | 2481 | 100.00% | 86.86% | SAMEA5042986 |
| C068UR | 54396 | 17456 | 15352 | 100.00% | 86.72% | SAMEA5042937 |
| T000UR | 57607 | 11995 | 9166 | 100.00% | 37.51% | SAMEA5043045 |
| T001UR | 47924 | 13474 | 2849 | 100.00% | 48.58% | SAMEA5043014 |
| T002UR | 63839 | 18381 | 3623 | 100.00% | 49.71% | SAMEA5042975 |
| T003UR | 70242 | 19166 | 5255 | 100.00% | 51.67% | SAMEA5042988 |
| T004UR | 67280 | 20578 | 2947 | 100.00% | 57.24% | SAMEA5042943 |
| T005UR | 52820 | 12868 | 4931 | 100.00% | 57.92% | SAMEA5043019 |
| T006UR | 79409 | 19472 | 13710 | 100.00% | 19.58% | SAMEA5043017 |
| T007UR | 34173 | 1403 | 797 | 100.00% | 50.06% | SAMEA5042999 |
| T008UR | 30074 | 1346 | 1044 | 100.00% | 84.77% | SAMEA5043031 |



| Sample name | Sequencing amount | | | % of reads annotated to taxa | | Archive accession number |
|---|---|---|---|---|---|---|
| | #raw reads | #clean reads | #filtered reads | Genus | Species | |
| T009UR | 58440 | 10386 | 7936 | 100.00% | 81.92% | SAMEA5042950 |
| T010UR | 65382 | 17191 | 9801 | 100.00% | 47.54% | SAMEA5042967 |
| T011UR | 38550 | 1163 | 464 | 100.00% | 65.52% | SAMEA5043061 |
| T012UR | 75848 | 21956 | 4605 | 100.00% | 52.62% | SAMEA5043049 |
| T013UR | 26872 | 1383 | 203 | 100.00% | 74.38% | SAMEA5042996 |
| T014UR | 23298 | 1741 | 518 | 100.00% | 93.44% | SAMEA5043042 |
| T015UR | 40653 | 2052 | 1470 | 100.00% | 29.25% | SAMEA5042953 |
| T016UR | 58448 | 16261 | 3993 | 100.00% | 58.75% | SAMEA5043053 |
| T017UR | 58703 | 19270 | 1929 | 100.00% | 54.12% | SAMEA5042990 |
| T018UR | 54726 | 12668 | 7152 | 100.00% | 17.18% | SAMEA5042970 |
| T019UR | 67711 | 14153 | 1606 | 100.00% | 58.90% | SAMEA5042954 |
| T020UR | 89936 | 22579 | 17919 | 100.00% | 68.74% | SAMEA5043052 |
| T021UR | 66094 | 14761 | 5276 | 100.00% | 29.66% | SAMEA5042951 |
| T022UR | 28712 | 803 | 422 | 100.00% | 74.88% | SAMEA5042940 |
| T023UR | 27738 | 1338 | 385 | 100.00% | 30.65% | SAMEA5042961 |
| T024UR | 19345 | 824 | 55 | 100.00% | 90.91% | SAMEA5042948 |
| T025UR | 29739 | 1578 | 1214 | 100.00% | 76.85% | SAMEA5042995 |
| T026UR | 79923 | 17606 | 6269 | 100.00% | 37.09% | SAMEA5042959 |
| T027UR | 61145 | 10093 | 6169 | 100.00% | 59.51% | SAMEA5042949 |
| T028UR | 71755 | 19529 | 16118 | 100.00% | 7.61% | SAMEA5042965 |
| T029UR | 58776 | 11505 | 8300 | 100.00% | 6.04% | SAMEA5043018 |
| T030UR | 57098 | 7000 | 5119 | 100.00% | 70.23% | SAMEA5042966 |
| T031UR | 49283 | 16113 | 1636 | 100.00% | 38.63% | SAMEA5043063 |
| T032UR | 46822 | 15041 | 1187 | 100.00% | 53.24% | SAMEA5042927 |
| T033UR | 63044 | 14272 | 10097 | 100.00% | 80.89% | SAMEA5043043 |
| T035UR | 50618 | 12403 | 1122 | 100.00% | 76.20% | SAMEA5043002 |
| T036UR | 78781 | 22492 | 17075 | 100.00% | 84.36% | SAMEA5042982 |
| T038UR | 73752 | 15237 | 11784 | 100.00% | 1.14% | SAMEA5043022 |
| T039UR | 58904 | 22286 | 19836 | 100.00% | 95.82% | SAMEA5043015 |
| T040UR | 77039 | 15332 | 8059 | 100.00% | 40.76% | SAMEA5043028 |
| T041UR | 58382 | 13735 | 11893 | 100.00% | 97.26% | SAMEA5043056 |
| T042UR | 53948 | 17112 | 1392 | 100.00% | 29.74% | SAMEA5043026 |
| T043UR | 72662 | 15446 | 10584 | 100.00% | 16.18% | SAMEA5042956 |
| T044UR | 59818 | 19779 | 724 | 100.00% | 73.48% | SAMEA5042993 |
| T045UR | 63627 | 21438 | 19282 | 100.00% | 39.34% | SAMEA5043033 |
| T046UR | 58142 | 20606 | 912 | 100.00% | 56.69% | SAMEA5042991 |
| T047UR | 24190 | 736 | 26 | 100.00% | 73.08% | SAMEA5042978 |
| T048UR | 10255 | 441 | 53 | 100.00% | 79.25% | SAMEA5043036 |
| T049UR | 63640 | 22520 | 20236 | 100.00% | 98.73% | SAMEA5043010 |
| T051UR | 22322 | 1066 | 101 | 100.00% | 77.23% | SAMEA5042997 |
| T052UR | 57909 | 11757 | 6419 | 100.00% | 67.85% | SAMEA5043058 |
| T053UR | 63637 | 24574 | 21178 | 100.00% | 99.17% | SAMEA5043034 |
| T054UR | 59194 | 18986 | 17148 | 100.00% | 15.87% | SAMEA5042929 |
| T055UR | 70744 | 14983 | 2724 | 100.00% | 84.07% | SAMEA5042962 |
| T056UR | 58876 | 19486 | 846 | 100.00% | 65.84% | SAMEA5043051 |
| T057UR | 53598 | 13889 | 12456 | 100.00% | 96.76% | SAMEA5042939 |
| T058UR | 18302 | 739 | 45 | 100.00% | 64.44% | SAMEA5043044 |
| T059UR | 59974 | 15542 | 12378 | 100.00% | 23.53% | SAMEA5042964 |
| T060UR | 21736 | 500 | 203 | 100.00% | 89.16% | SAMEA5042946 |
| T061UR | 33002 | 1153 | 503 | 100.00% | 56.46% | SAMEA5043011 |
| T062UR | 64983 | 10519 | 6302 | 100.00% | 52.19% | SAMEA5042933 |
| T063UR | 53347 | 12793 | 4023 | 100.00% | 40.00% | SAMEA5043041 |

Table 1. (Continued)

| Table 1. (Continued) | | | | | | |
|---|---|---|---|---|---|---|
| Sample name | Sequencing amount | | | % of reads annotated to taxa | | Archive accession number |
| | #raw reads | #clean reads | #filtered reads | Genus | Species | |
| T064UR | 68122 | 23665 | 21094 | 100.00% | 79.99% | SAMEA5042934 |
| T065UR | 51210 | 17070 | 1242 | 100.00% | 65.30% | SAMEA5042977 |
| T066UR | 64589 | 26532 | 1911 | 100.00% | 56.04% | SAMEA5042968 |
| T067UR | 67938 | 16248 | 4974 | 100.00% | 50.12% | SAMEA5043005 |
| T068UR | 70192 | 28890 | 3698 | 100.00% | 40.37% | SAMEA5043013 |
| T069UR | 60564 | 21236 | 17683 | 100.00% | 43.97% | SAMEA5042957 |
| T070UR | 83453 | 20755 | 5034 | 100.00% | 44.18% | SAMEA5043038 |
| T071UR | 80077 | 36770 | 29224 | 100.00% | 97.68% | SAMEA5043007 |
| T072UR | 73469 | 29671 | 19787 | 100.00% | 87.68% | SAMEA5042992 |
| T073UR | 73167 | 17577 | 3914 | 100.00% | 54.09% | SAMEA5042989 |
| T074UR | 59084 | 23906 | 21347 | 100.00% | 89.11% | SAMEA5042973 |
| T075UR | 60263 | 17726 | 15250 | 100.00% | 70.37% | SAMEA5042976 |
| T076UR | 37428 | 809 | 514 | 100.00% | 63.42% | SAMEA5042960 |
| T078UR | 76834 | 17034 | 4220 | 100.00% | 66.75% | SAMEA5042958 |
| T080UR | 12172 | 609 | 61 | 100.00% | 52.46% | SAMEA5042932 |
| T081UR | 63432 | 14841 | 6915 | 100.00% | 86.49% | SAMEA5043029 |
| T082UR | 26941 | 693 | 609 | 100.00% | 2.96% | SAMEA5043047 |
| T083UR | 69149 | 34307 | 30270 | 100.00% | 95.45% | SAMEA5043055 |
| T084UR | 59304 | 25863 | 22866 | 100.00% | 89.34% | SAMEA5042928 |
| T085UR | 65565 | 20426 | 1344 | 100.00% | 78.57% | SAMEA5043030 |
| T086UR | 66605 | 23828 | 21243 | 100.00% | 95.16% | SAMEA5043025 |
| T087UR | 62480 | 16656 | 6414 | 100.00% | 76.55% | SAMEA5043057 |
| T088UR | 82733 | 32538 | 3794 | 100.00% | 71.09% | SAMEA5042994 |
| T089UR | 110227 | 27223 | 11761 | 100.00% | 24.49% | SAMEA5042944 |
| T090UR | 70526 | 29296 | 1917 | 100.00% | 71.62% | SAMEA5043020 |
| T091UR | 27973 | 913 | 739 | 100.00% | 27.74% | SAMEA5042952 |
| T092UR | 69694 | 10825 | 7894 | 100.00% | 55.26% | SAMEA5043050 |
| T093UR | 58492 | 14656 | 7272 | 100.00% | 84.27% | SAMEA5042980 |
| T094UR | 194564 | 59268 | 35224 | 100.00% | 37.26% | SAMEA5042971 |
| T095UR | 42681 | 1009 | 560 | 100.00% | 45.00% | SAMEA5042974 |

pre-processed for quality control and filtering, then clustered into OTUs (Methods, Table 1 and **OTU_table_urine.biom.hdf5** [8]).

Due to anatomical structures, voided urine samples from women were considered to be easily contaminated by microbiota from the surrounding vulvovaginal region [22]. Most vaginal communities (88%) in this cohort were dominated by one genus with >50% relative abundance within data from individuals. In contrast, the urinary microbiota in this study showed more heterogeneity. 56.93% of the cohort harboured a diverse type represented significantly by bacteria, including *Streptococcus, Lactobacillus, Pseudomonas, Staphylococcus, Acinetobacter,* and *Vagococcus*, though none of these species were dominant, i.e. reached >50% relative abundance (Figure 2). In addition, 22.63% of the women harboured >50% *Streptococcus*, and 13.87% of the women harboured >50% *Lactobacillus* (Figure 2A, B). Rare subtypes such as *Enterococcus* (2.19%), *Bifidobacteriaceae* (1.46%), *Prevotella* (0.73%), *Enterobacteriaceae* (0.73%), *Coriobacteriaceae* (0.73%), and *Veillonella* (0.73%) were also detected in this cohort (Figure 2A, B). Notably, the median relative abundances of *Lactobacillus, Pseudomonas,* and *Acinetobacter* in the urine samples were more similar to the uterus samples (Figure 2C) [2]. At the phylum level, urinary microbiota were dominated by Firmicutes and Proteobacteria (Figure 2C).



**Figure 2.** Urinary microbiota of the initial cohort of 137 Chinese reproductive-age women. (A) The relative abundances of genera detected in each individual are shown in the bar chart. The dendrogram is a result of a centroid linkage hierarchical clustering based on Euclidean distances between the microbial composition proportion of urinary bacterial communities. (B) The ratio of different urinary microbiota types. The genus whose relative abundance accounted for >50% in an individual was selected as an identified type. The genera that accounted for <50% of the microbiota in an individual were identified as diverse type. (C) Pie chart for the urinary microbial genera according to their median relative abundance. Genera that took up less than 1% of the microbiota are labelled together as 'others'. The outer ring indicates the distribution of microbiota at the phylum level.

## Cultivation of live bacteria from transurethral catheterized urine

The question of whether bacterial DNA signals have originated from live bacteria or fragments in the urine samples has been a subject of much debate [22]. To demonstrate the utility of the data for addressing this question, we performed a validation study using live bacteria cultures from urine samples provided by an additional cohort of 10 women.

**Table 2.** Identification of cultured microbial isolates from urine of the 10 additional women by sequencing of partial 16S rRNA gene.

| Sample ID | Condition | Medium | 16S rRNA gene-PCR Identification | Accessions | Identity (%) | Supported by previous cultivation |
|---|---|---|---|---|---|---|
| S001U | Anaerobic, 37 ˚C | EG | *Clostridium cochlearium* | LR761333.1 | 99.26 | Meijer-Severs *et al.* [24] |
| S001U | Anaerobic, 37 ˚C | 104 | *Streptococcus sp.* (*S. tigurinus/S. mitis*) | LR761334.1 | 99.72 | Hilt *et al.* [23] |
| S003U | Anaerobic, 37 ˚C | BHI | *Enterococcus faecalis* | LR761335.1 | 99.91 | Hilt *et al.* [23], Guzmàn *et al.* [25], Fraimow *et al.* [26], |
| S003U | Anaerobic, 37 ˚C | 104 | *Lactobacillus crispatus* | LR761337.1 | 99.82 | Hilt *et al.* [23] |
| S003U | Anaerobic, 37 ˚C | 104 | *Propionibacterium granulosum* | LR761336.1 | 99.02 | Ormerod *et al.* [27] |
| S008U | Anaerobic, 37 ˚C | 104, BHI, EG | *Streptococcus agalactiae* | LR761340.1, LR761339.1, LR761338.1 | 99.65, 99.35, 99.52 | Hilt *et al.* [23] |

We tried to culture and isolate bacterial colonies from freshly collected urine samples. Urine samples were serial diluted and spread on three different kinds of agar plates and incubated under both aerobic and anaerobic conditions. Six different positive isolates belonging to 5 genera, including *Lactobacillus*, *Staphylococcus*, *Clostridium*, *Enterococcus*, and *Propionibacterium* were obtained from 3 out of 10 subjects (Table 2). The 5 genera were also found as dominant in our 16S rRNA gene amplicon sequencing data and consistent with previous cultivation results of published papers [23–26] (Table 2). Reassuringly, no isolates were detected from the negative controls (sterile saline and ultrapure water). Therefore, these data verified the existence of live bacteria in the urine by obtaining isolates using conventional culturing methods.

## Considerable bacterial biomass revealed by qPCR

To provide additional evidence of the bacterial communities in the urine, a species-specific quantitative real-time PCR method was utilized to focus on the four common vaginal *Lactobacillus* species, i.e. *L. crispatus*, *L. iners*, *L. jensenii* and *L. gasseri* (**QPCR Lactobacillus.csv** [8]). The *Lactobacillus* species we examined presented a similar distribution and abundance along the female reproductive tract, and the corresponding urinary *Lactobacillus* ranged between the upper and lower reproductive tracts (Figure 3A). Among them, *L. iners* occurred most frequently (59%) in the urine samples, while *L. crispatus* only occurred in 26% of women sampled (Figure 3B). *L. iners* was reported far less protective against bacterial and viral infections compared to *L. crispatus* [28]. 80% of the cohort was detected to harbour at least one of these four *Lactobacillus* species (Figure 3B). The occurrence rate of *Lactobacillus* in the genus level of 16S rRNA gene amplicon sequencing data was 94% (Figure 2A). The total bacterial biomass is approximated by the ratio of the copy number from the result of qPCR to the relative abundance according to the result of 16S rRNA gene sequencing of the same sample (**QPCR bacterial_biomass.csv** [8]). The result gave an estimation of $10^7$ copies/sample, placing the urinary bacterial biomass between the vaginal-cervical sites ($10^{10}$–$10^{11}$ copies/sample) and the endometrium (ET) samples ($10^6$–$10^7$ copies/sample) [2] (Figure 3A), all of which were orders of magnitude above potential background noise [29]. These results were interestingly consistent with a weakly acidic pH of the urine, in comparison to pH < 4.5 in the vagina or pH ~ 8 in the peritoneal fluid [30].

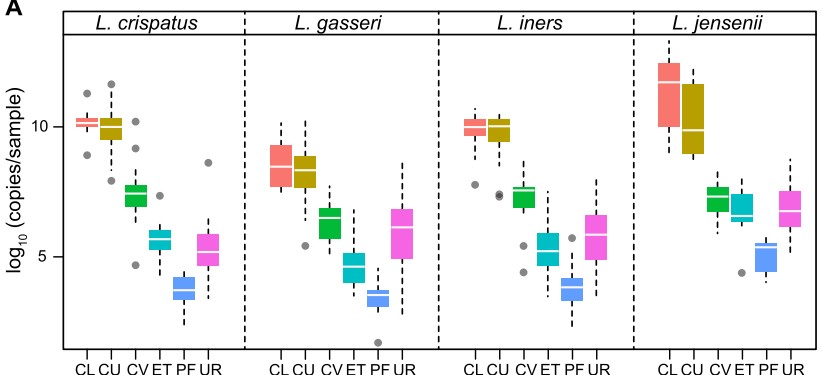
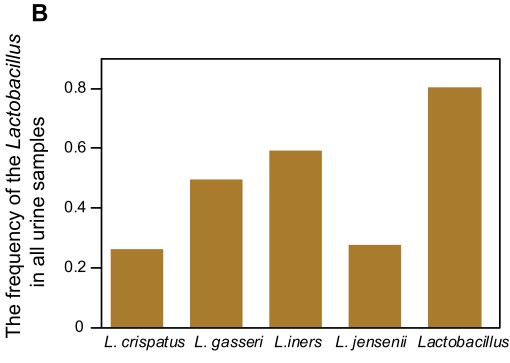

**Figure 3.** The concentrations of the dominant _Lactobacillus_ species at urine and the reproductive tract. Samples derive from the initial cohort of 137 Chinese reproductive-age women. (A) The abundance of _L. iners, L. jensenii, L. crispatus_ and _L. gasseri_ calculated by qPCR results in different samples. Boxes denote the interquartile range (IQR) between the first and third quartiles (25th and 75th percentiles, respectively), and the lines inside the boxes denote the median. The whiskers denote the lowest and highest values within 1.5 times the IQR from the first and third quartiles, respectively. (B) The frequency of the respective _Lactobacillus_ detected in all urine sample.

## Intra-individual similarity in the urine-reproductive tract microbiota

To further assess the microbiota relationship between the urine and the six positions of the female reproductive tract, we computed intra-individual correlation between the microbial profiles in the urine and those found in different sites of the reproductive tract, and then clustered the individuals into 4 groups (Spearman's correlation coefficient, Figure 4A, **relative_abundance_correlation.csv** [8]). Interestingly, the microbiota of group 3, which accounted for 41% of the cohort, showed significant correlation between the urine samples and the female reproductive tract samples, of which the coefficient increased gradually along the anatomical site from CL to CV, ET, and PF (Figure 4B). In contrast, 9% of women in group 1 presented a reverse trend. In group 2 (22%) and group 4 (27%), there appeared to be a weak relationship between the microbiota of the urine and female reproductive tract. Taken together, we observed the most similar distribution of microbiota between urine and CV/ET (Figure 4A). The principal coordinate analyses (PCoA) of the weighted and unweighted intra-individual UniFrac distance further corroborated our conclusion that there is an intra-individual similarity of the microbiota between the urine and the upper sites of female reproductive tract, especially the junction sites (CV and ET) (Figure 5).

## Lifestyle and clinical factors influencing the urinary microbiota

The human microbiome is dynamic and highly affected by its host environment. Age, menstrual cycle, benign conditions such as adenomyosis, and infertility due to endometriosis have previously been reported to shape the vagino-uterine microbiota [2]. With our comprehensive collection of demographic and baseline clinical characteristics from women of reproductive age (**sample_metadata.csv** [8]), such variations in the urinary microbiota can be explored in this dataset. Urinary microbial composition was significantly associated with these factors, such as age, surgical history, abortion, vaginal deliveries, experience of given birth (multipara vs. nullipara), infertility due to endometriosis, and hysteromyoma (PERMANOVA, _P_ <0.05, _q_ <0.05, Table 3). Although the urinary microbiota also correlated with some other factors, such as menstrual phase,



**A**

Spearman's correlation coefficient

0 0.1 0.2 0.3 0.4 0.5 0.6 0.7

Urinary microbiota types

- Bifidobacteriaceae
- Coriobacteriaceae
- Prevotella
- Enterococcus
- Lactobacillus
- Streptococcus
- Veillonella
- Diverse

**B**

**Figure 4.** Similarity of the urine-reproductive tract microbiota within individuals. (A) Heatmap for the intra-individual Spearman's correlation coefficient between microbiota identified in the urine and at different sites in the reproductive tract (**relative_abundance_correlation.csv** [8]). Samples derived from the initial cohort of 95 Chinese reproductive-age women, who collected both the urine and reproductive tract samples. As the number of samples from fallopian tubes (FLL, FRL) is too small, the correlation between microbiota in the urine and those in fallopian tubes are not shown. The dendrogram is a result of a centroid linkage hierarchical clustering based on Euclidean distances between the intra-individual Spearman's correlation coefficient of different body sites. The colored squares illustrate the subtypes found within the urinary microbiome. (B) Spearman's correlation coefficient between microbiota found in the urine and those from different sites of the reproductive tract. The Wilcoxon ranked sum test was used to calculate the difference. Boxes denote the interquartile range (IQR) between the first and third quartiles (25th and 75th percentiles, respectively), and the line inside the boxes denote the median. The whiskers denote the lowest and highest values within 1.5 times the IQR from the first and third quartiles, respectively. An asterisk denotes $p < 0.05$, two asterisks denote $p < 0.01$, three asterisks denote $p < 0.001$.

contraception, endometriosis, pelvic adhesiolysis, and anemia, statistical significance was not achieved after controlling for multiple testing (PERMANOVA, $P < 0.05$ but $q > 0.05$,





**Figure 5.** PCoA on the samples based on Unweighted-UniFrac (A) and Weighted-UniFrac (B) distances. Samples were taken from UR, CL, CU, and CV before operation, and from ET and PF during operation. Samples were derived from the initial cohort of 137 Chinese reproductive-age women. Each dot represents one sample ($n$ =94 CL, 95 CU, 95 CV, 80 ET, 93 PF, 9 FLL, 10 FRL, and 137 UR).

Table 3). The initial results here indicate a close link between the urinary microbiota and the general and diseased physiological conditions, and this link could be further understood by exploring this data more deeply.

## Potential uses

As a large-scale cohort for studying the female urinary microbiota, our data provide a useful baseline and reference dataset in women of reproductive age. We also explored the association between the composition of urinary microbiota and that of the female reproductive tract microbiota. It is valuable to note that a higher intra-individual

**Table 3.** PERMANOVA for the influence of phenotypes on the urinary microbiota.

| Phenotype | Bray-Curtis | | | Unweighted-UniFrac | | | Weighted-UniFrac | | |
|---|---|---|---|---|---|---|---|---|---|
| | R2 | *P* value | Fdr | R2 | *P* value | Fdr | R2 | *P* value | Fdr |
| Age | 0.018 | 0.005 | 0.049 | 0.010 | 0.178 | 0.541 | 0.019 | 0.026 | 0.419 |
| Age-2 groups | 0.013 | 0.050 | 0.236 | 0.011 | 0.116 | 0.429 | 0.016 | 0.042 | 0.452 |
| Age-3 groups | 0.032 | 0.026 | 0.150 | 0.025 | 0.228 | 0.577 | 0.030 | 0.135 | 0.539 |
| Pulses | 0.015 | 0.019 | 0.131 | 0.010 | 0.159 | 0.505 | 0.015 | 0.072 | 0.456 |
| Frequent colds | 0.011 | 0.080 | 0.270 | 0.011 | 0.108 | 0.429 | 0.021 | 0.017 | 0.419 |
| Antibiotics | 0.014 | 0.036 | 0.194 | 0.012 | 0.108 | 0.429 | 0.007 | 0.525 | 0.782 |
| Constipation | 0.011 | 0.114 | 0.325 | 0.014 | 0.039 | 0.400 | 0.018 | 0.033 | 0.419 |
| Surgical history | 0.018 | 0.005 | 0.049 | 0.018 | 0.006 | 0.172 | 0.034 | 0.001 | 0.091 |
| Abdominal surgical history | 0.010 | 0.187 | 0.418 | 0.007 | 0.466 | 0.755 | 0.019 | 0.030 | 0.419 |
| Menstrual cycle | 0.009 | 0.200 | 0.421 | 0.018 | 0.005 | 0.172 | 0.015 | 0.065 | 0.455 |
| Menstrual phase (lower) | 0.018 | 0.260 | 0.468 | 0.024 | 0.048 | 0.408 | 0.020 | 0.207 | 0.623 |
| Menstrual phase (upper) | 0.018 | 0.006 | 0.056 | 0.018 | 0.009 | 0.172 | 0.014 | 0.096 | 0.456 |
| Contraception | 0.044 | 0.010 | 0.086 | 0.038 | 0.098 | 0.429 | 0.029 | 0.470 | 0.782 |
| Vaginal deliveries | 0.018 | 0.003 | 0.049 | 0.016 | 0.014 | 0.172 | 0.016 | 0.051 | 0.455 |
| Abortion | 0.028 | 0.004 | 0.049 | 0.014 | 0.239 | 0.585 | 0.016 | 0.198 | 0.623 |
| Multipara / nullipara | 0.019 | 0.003 | 0.049 | 0.017 | 0.014 | 0.172 | 0.013 | 0.091 | 0.456 |
| Infertility due to endometriosis | 0.045 | 0.000 | 0.008 | 0.029 | 0.013 | 0.172 | 0.019 | 0.181 | 0.599 |
| Endometriosis | 0.014 | 0.022 | 0.141 | 0.011 | 0.095 | 0.429 | 0.005 | 0.644 | 0.857 |
| Pelvic adhesiolysis | 0.008 | 0.346 | 0.572 | 0.013 | 0.042 | 0.400 | 0.006 | 0.489 | 0.782 |
| Anemia | 0.016 | 0.012 | 0.090 | 0.008 | 0.354 | 0.740 | 0.006 | 0.511 | 0.782 |
| Hysteromyoma | 0.018 | 0.003 | 0.049 | 0.012 | 0.057 | 0.429 | 0.021 | 0.014 | 0.419 |

compositional similarity was observed between the microbiota of the urine and those of the cervical canal/uterus than between the microbiota of the urine and those of the vagina. This finding indicates that sampling of midstream urine (the least invasive and the easiest way) could be potentially used to survey the micro-environment of the cervical canal and uterus in the general population. This is relevant to the demonstrated associations between the urinary microbiota and various uterine-related diseases, such as hysteromyoma and infertility due to endometriosis. Our data provide a reference for clinical diagnosis and warrants further detailed exploration.

There are three limitations for this study. Firstly, as it was not possible to directly sample the upper reproductive tract of perfectly healthy women, we have included women who underwent minimally invasive laparoscopy or laparotomy for conditions that are not known to involve infection. This was the best proxy for sampling the upper reproductive tract in healthy women. Nevertheless, the relevance of the urinary microbiota between healthy women and women in our cohort would require further comparison. Secondly, for the low bacterial biomass of urine samples, a more comprehensive sampling process should be taken into consideration in subsequent studies, such as disinfection of the urethra and vulvovaginal region with 75% alcohol before urine self-collection, including a sample of sterile saline with the self-collection kit as a negative control and asking participants to fill another vial with it immediately following urine collection. A comparison of the microbial composition between the catheter-collected and self-collected specimens in the same individual would also require further inspection. Together, we hope that this dataset helps promote a new round of accelerated discoveries, including a novel scientific explanation for uterine-related diseases via longitudinal studies on the microbiota of the urinary and reproductive tracts.



## DECLARATIONS
## ETHICS APPROVAL AND CONSENT TO PARTICIPATE

The study was approved by the Institutional Review Board of BGI-Shenzhen (No. BGI-IRB 17219) and Peking University Shenzhen Hospital (Version 1.0.20140301). All participants gave written informed consent prior to their recruitment into the study.

## DATA AVAILABILITY

The sequence reads generated by 16S rRNA gene amplicon sequencing have been deposited in both the European Nucleotide Archive with the accession number PRJEB29341 and the CNSA (https://db.cngb.org/cnsa/) of CNGB database with accession code CNP0000166. Additional data, result and a STORMS (Strengthening The Organizing and Reporting of Microbiome Studies) checklist are available from the *GigaScience* GigaDB repository [8]. The sequences of bacterial isolates have been deposited in the European Nucleotide Archive with the accession number PRJEB36743.

## AUTHOR CONTRIBUTIONS

H.J. and R.W. organized this study. W.W., J.D., H.D., L.Z., H.T., T.W., and R.W. performed the sample collection, and phenotypic information collection. F.L., L.S., C.C., and J.L. performed the molecular biology experiments. C.C., L.H., and F.L. performed the bioinformatic analyses. C.C., X.Z., F.L., and H.J., wrote the manuscript.

## COMPETING INTERESTS

There were no competing financial interests.

## ACKNOWLEDGEMENTS

The study was supported by the Shenzhen Municipal Government (No. SZXK027 and No. SZSM202011016), Shenzhen Peacock Plan (No. KQTD20150330171505310), and the Medical Scientific Research Foundation of Guangdong (No. A2019035). The authors really appreciate colleagues at BGI-Shenzhen for DNA extraction, library construction, and sequencing.

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
