## [Reviewer Report]

Reviewer name and names of any other individual's who aided in reviewer Chris HunterDo you understand and agree to our policy of having open and named reviews, and having your review included with the published papers. (If no, please inform the editor that you cannot review this manuscript.)YesIs the language of sufficient quality?YesPlease add additional comments on language quality to clarify if needed
Are all data available and do they match the descriptions in the paper? NoAdditional Commentsline 96-97 
"In this study, a total of 147 reproductive age women (age 22-48) were recruited by Peking University Shenzhen Hospital (Supplementary Table 1)."
B utSup. table 1 has only 137 samples.
Revise text to explain only 137 samples were used for the main analysis, with the 10 extra for validation.

Line 103 -104
"None of the subjects received any hormone treatments, antibiotics or vaginal medications within a month of sampling."
Sup Table 1 has a column for "Antibiotic use True/False", 41 samples have "T"? this needs explaining. Its possible the spreadsheet True is referring to a longer time period, but thats not explained anywhere.

line 110-112
"The samples from an additional 10 women were collected for validation purposes by a doctor during the surgery in July 2017."
Where are these metadata? they are not included in Sup table 1.

The data presented and discussed in "additional-findings.docx" are not included in the data files (yet), these should either be removed (as not included in the main article), or expand upon the methods (to include negative control details) and add this to main text.Are the data and metadata consistent with relevant minimum information or reporting standards? See GigaDB checklists for examples <a href="http://gigadb.org/site/guide" target="_blank">http://gigadb.org/site/guide</a>YesAdditional CommentsThe supplemental tables need some better legends/descriptions to help readers understand what data is in them.Is the data acquisition clear, complete and methodologically sound?YesAdditional CommentsThe wet and bioinformatics methods could benefit from being included in protocols.io Is there sufficient detail in the methods and data-processing steps to allow reproduction?YesAdditional CommentsIs there sufficient data validation and statistical analyses of data quality? YesAdditional CommentsIs the validation suitable for this type of data?YesAdditional CommentsIs there sufficient information for others to reuse this dataset or integrate it with other data?YesAdditional CommentsThe Figure appear to be mixed up, whats displayed as Figure 1 in the manuscript appears to relate to the legend given for Figure 2, Figure 2 relates to legend of Figure 3, and Figure 3 relates to the legend of Fig 1!!!

line 69 -Chen et al. no citation number link provided
line 74 -Thomas-White et al. (2018) no citation number link provided
line 79 -Gottschick et al. (2017) no citation number link provided

line 246-248
"The initial results here indicate a close link between the urinary microbiota with the general and diseased physiological conditions,... "
As this study is looking at "Healthy" individuals I do not believe there is sufficient evidence to back up this statement about the "diseased" physiological conditions.

line 274-275 
"The sequences of bacterial isolates have been deposited in the European Nucleotide Archive with the accession number PRJEB36743" 
this accession is not public so I am unable to see whats included here.

If available we would like to see the Real-Time PCR Data from the experiments made available in Real-Time PCR Data Markup Language (RDML).

The additional cohort of 10 women is almost a different study, it didn't have the same 16s RNA amplicon sequencing done, and was only a validation that some live bacteria can be cultured from urine in a small number of cases (3/10). If it is to be included table S5 should be updated to include the specific INSDC accessions for the submitted sequences. (title of Table S5 in file is currently saying Table 1)Any Additional Overall Comments to the AuthorRecommendationMinor Revision

---

## [Reviewer Report]

Upload additional filesDRR-20201006/form/Urine_cc20200910_review.docxReviewer name and names of any other individual's who aided in reviewer levi WaldronDo you understand and agree to our policy of having open and named reviews, and having your review included with the published papers. (If no, please inform the editor that you cannot review this manuscript.)YesIs the language of sufficient quality?YesPlease add additional comments on language quality to clarify if needed
Are all data available and do they match the descriptions in the paper? YesAdditional CommentsAre the data and metadata consistent with relevant minimum information or reporting standards? See GigaDB checklists for examples <a href="http://gigadb.org/site/guide" target="_blank">http://gigadb.org/site/guide</a>NoAdditional CommentsSee reportIs the data acquisition clear, complete and methodologically sound?YesAdditional CommentsIs there sufficient detail in the methods and data-processing steps to allow reproduction?YesAdditional CommentsIs there sufficient data validation and statistical analyses of data quality? YesAdditional CommentsIs the validation suitable for this type of data?YesAdditional CommentsIs there sufficient information for others to reuse this dataset or integrate it with other data?YesAdditional CommentsSee uploaded report of review.Any Additional Overall Comments to the AuthorRecommendationMajor Revision

---

## [Editor Report]

Comments to the AuthorThanks again for your submission, and we are very happy with the revisions now.